# Single-crystal ZrCo nanoparticle for advanced hydrogen and H-isotope storage

Zhenyang Li[1,4], Shiyuan Liu[1,4], Yanhui Pu[1], Gang Huang[1], Yingbo Yuan[1], Ruiqi Zhu[1], Xufeng Li[1], Chunyan Chen[1], Gao Deng[1], Haihan Zou[1], Peng Yi[1], Ming Fang[1], Xin Sun[2], Junzhe He[1,2], He Cai[2], Jiaxiang Shang[1], Xiaofang Liu[1] ✉, Ronghai Yu[1] ✉ & Jianglan Shui[1,3] ✉

Hydrogen-isotope storage materials are essential for the controlled nuclear fusion. However, the currently used smelting-ZrCo alloy suffers from rapid degradation of performance due to severe disproportionation. Here, we reveal a defect-derived disproportionation mechanism and report a nano-single-crystal strategy to solve ZrCo's problems. Single-crystal nano-ZrCo is synthesized by a wet-chemistry method and exhibits excellent comprehensive hydrogen-isotope storage performances, including ultrafast uptake/release kinetics, high anti-disproportionation ability, and stable cycling, far superior to conventional smelting-ZrCo. Especially, a further incorporation of Ti into nano-ZrCo can almost suppress the disproportionation reaction. Moreover, a mathematical relationship between dehydrogenation temperature and ZrCo particle size is established. Additionally, a microwave method capable of nondestructively detecting the hydrogen storage state of ZrCo is developed. The proposed disproportionation mechanism and anti-disproportionation strategy will be instructive for other materials with similar problems.

Controlled nuclear fusion of hydrogen isotopes promises to provide society with limitless clean and dense energy[1,2]. The International Thermonuclear Experimental Reactor project (ITER) was launched in 1985 with the aim of developing technologies for the safe use of nuclear fusion energy, and has achieved remarkable progress[3–6]. For example, sustained energy pulses were realized using equal amounts of tritium and deuterium fuels[7]. Recently, another exciting success in ignition has also been reported by US Department of Energy[8]. Hydrogen-isotope storage material is an essential component of the system and controls the delivery of fuels[9–11]. Depleted uranium has long been used for hydrogen-isotope storage, but the drawbacks such as spontaneous combustion, radioactivity, scarcity, and high cost limit its application[12–16]. In contrast, ZrCo intermetallic compound has the advantages of safety, the higher storage capacity, and excellent ³He trapping ability (facilitate obtaining high purity fuel)[17–20], and are currently the only substitute for uranium.

ZrCo powders are traditionally prepared by smelting and casting methods followed by mechanical milling to micron size (denoted as smelting-ZrCo), and have been used on a small scale[21]. Smelting-ZrCo suffers from pulverization and severe disproportionation reaction ($2ZrCoX_3 \rightarrow ZrCo_2 + ZrX_2 + 2X_2$, X = H, D, T) that produces a large number of useless components of $ZrCo_2$ and $ZrX_2$ in just several cycles. Moreover, smelting-ZrCo usually requires dozens of hours of activation before use. These problems lead to a significant decline of cyclic storage capacity and a waste of precious H-isotope (e.g. ~ $30,000 g^{-1}$ tritium)[3,22,23]. The 8e site of the ZrCo crystal is conventionally regarded as the origin of disproportionation. Doping ZrCo with elements such as Ti, Nb and Hf can reduce disproportionation by decreasing the lattice size of the 8e site[24–29]. However, this doping strategy usually reduces hydrogen storage capacity and has limited improvement in cyclability (from 0.4 wt.% to 1.2 wt.%)[27,29–31]. And the activation process of these alloys still lasts for

[1]School of Materials Science and Engineering, Beihang University, Xueyuan Road No.37, Beijing 100191, P. R. China. [2]National Key Laboratory of Scattering and Radiation, Beijing 100854, P. R. China. [3]Tianmushan Laboratory, Xixi Octagon City, Yuhang District, Hangzhou 310023, P. R. China. [4]These authors contributed equally: Zhenyang Li, Shiyuan Liu. ✉e-mail: liuxf05@buaa.edu.cn; rhyu@buaa.edu.cn; shuijianglan@buaa.edu.cn

several hours. This year, Chen's group[32,33] converted cubic-structured ZrCo into orthorhombic-structured $Zr_{0.8}Nb_{0.2}Co_{0.6}Cu_{0.15}Ni_{0.25}$ alloy through precise multicomponent substitution. This quinary alloy underwent isostructural phase transition with orthorhombic structure during de-/hydrogenation process, which can reduce lattice expansion and atomic motion, thereby achieving higher cycling stability. This study inspires researchers to reveal the disproportionation mechanism of ZrCo from different perspectives, and develop more strategies to improve the comprehensive performances of ZrCo alloys.

Here, we reveal that structural defects are associated with the disproportionation of ZrCo alloy, and consequently propose a solution of nano-single crystal to the disproportionation problem. Structural defects can act as "zippers" to induce and promote the disproportionation reaction. Owning to the low defect density, single-crystal ZrCo nanoparticles show significantly improved resistance to disproportionation. When conventional polycrystalline ZrCo microparticles reach 100% disproportionation at 500 °C within 300 min, the disproportionation ratio of single-crystal ZrCo nanoparticles is only 24%, which leads to significantly improved cycling stability. Meanwhile, single-crystal ZrCo nanoparticles have a higher H-isotope storage capacity (reaching the theoretical limit) and an order of magnitude higher kinetics than smelting-ZrCo. If Ti element is incorporated, single-crystal $Zr_{0.9}Ti_{0.1}Co$ nanoparticles (denoted as ZTC) can further reduce the disproportionation ratio to 5% while maintaining ultrafast kinetics and high cyclability. Moreover, a mathematical relationship is established to reveal the effect of particle size on the dehydrogenation thermodynamics. Additionally, a microwave method is developed to nondestructively detect the storage state of ZrCo. The reported disproportionation mechanism, anti-disproportionation strategy, and detection method are expected to have a profound impact on other hydrogen storage[34–36] and energy storage materials[37–39] with disproportionation problems.

## Results and discussion
### Material preparation, composition, and morphology
A "bottom-up" wet chemical method was developed to synthesize nanoscale ZrCo alloy (Supplementary Fig. 1a). Briefly, zirconium and cobalt ions were co-precipitated from the solution to produce zirconium-cobalt hydroxides (Supplementary Fig. 2), which were then dried and calcined in air to form oxides (Supplementary Fig. 3). Finally, ZrCo alloy particles were obtained by magnesiothermic reduction of oxides and pickling to remove MgO and residual Mg. The heat treatment temperatures of oxidation and magnesiothermic reduction

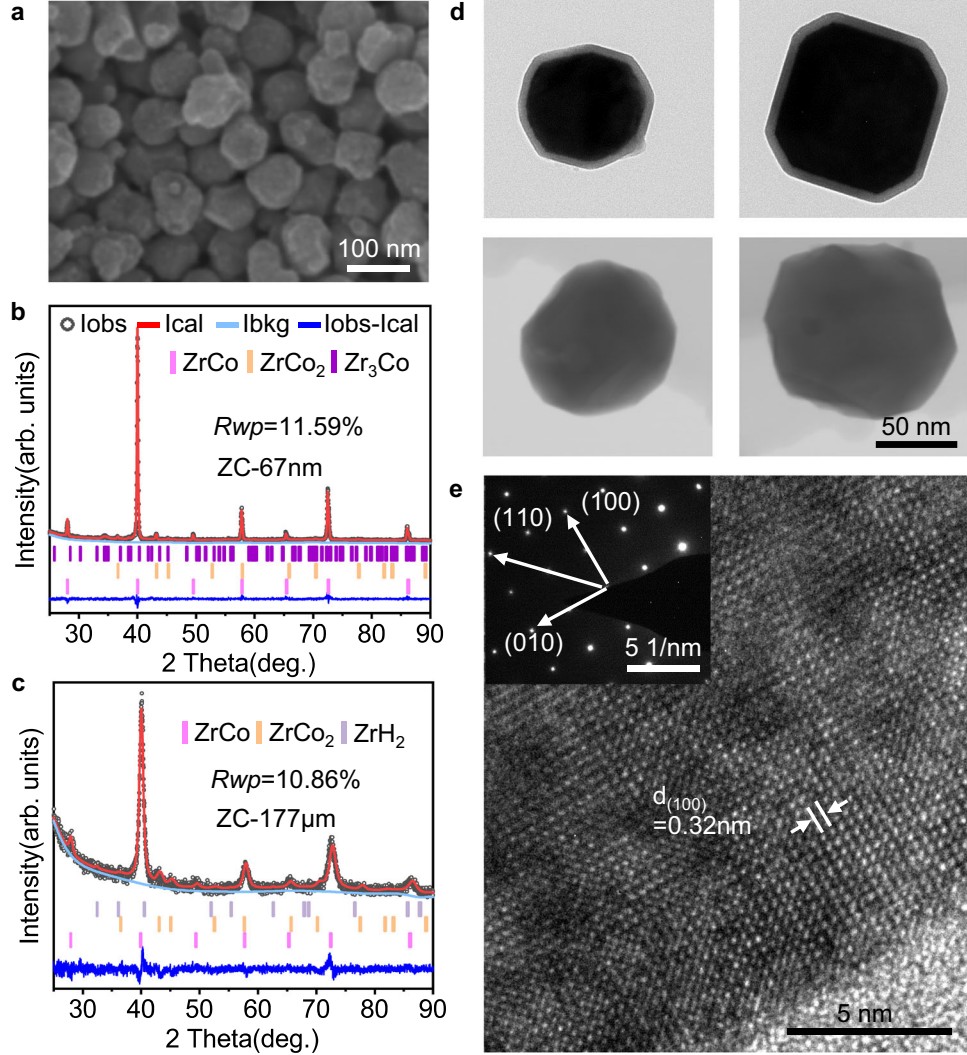

**Fig. 1 | Structural characteristics of ZrCo alloy. a** SEM and (**b**, **c**) XRD patterns and Rietveld refinement results of ZC-67nm and ZC-177μm. **d** TEM and STEM images of ZC-67nm. **e** High-resolution TEM image of ZC-67nm. The distance is the (100) crystal plane spacing of ZrCo (denote as $d_{(100)}$). Inset: corresponding SAED pattern.

**Table 1 | Rietveld refinement results of XRD patterns of ZC-67nm and ZC-177μm**

| Sample | Lattice constant of ZrCo phase (Å) | Cell volume of ZrCo phase (Å³) | Phase abundance (wt.%) | | | |
|---|---|---|---|---|---|---|
| | | | ZrCo | ZrCo$_2$ | Zr$_3$Co | ZrH$_2$ |
| ZC-67nm | 3.195 | 32.605 | 95.5 | 3.3 | 1.2 | 0 |
| ZC-177μm | 3.203 | 32.860 | 90.3 | 8.1 | 0 | 1.6 |

influence the crystallinity and purity of the ZrCo nanoparticles, and have been optimized in Supplementary Fig. 4. The particle size of ZrCo alloy can be adjusted from 67, 336, to 621 nm by changing the solution concentration and using freeze-drying technique; while the ZrCo particles subjected to thermal drying increases to 1.3 μm (Fig. 1a and Supplementary Figs. 5 and 6). These chemically-synthesized ZrCo particles (denoted as chem-ZrCo) have a quasi-spherical shape and uniform distribution of Zr and Co elements (Supplementary Figs. 7–10). Larger-sized ZrCo alloys in micron scale were prepared by conventional smelting method followed by mechanical crushing and sieving to obtain average particle sizes of ~42, 94, and 177 μm (Supplementary Fig. 1b). These smelting-ZrCo particles show irregular shape and relatively wide size distribution (Supplementary Fig. 11). All the above ZrCo alloys are denoted as ZC-n (n is the average particle size).

The X-ray diffraction (XRD) patterns are shown in Supplementary Fig. 12. All samples contain ZrCo as the main phase along with a small fraction of impurities. According to the Rietveld refinement results (Fig. 1b and Table 1), ZC-67nm is composed of 95.5% ZrCo, 3.3% ZrCo$_2$ and 1.2% Zr$_3$Co. Zr$_3$Co will convert into ZrCo$_2$ and ZrH$_2$ phases during the hydrogen absorption-desorption cycles[40,41]. The crystallite size of ZC-67nm was calculated to be 50.1 nm using the Rietveld refinement method, which is close to the particle size observed in transmission electron microscopy (TEM) and scanning transmission electron microscopy (STEM) images (Fig. 1d). This implies that ZC-67nm particles are almost single crystals. Selected area electron diffraction (SAED) and high-resolution TEM (HRTEM) images further confirm the single-crystalline characteristics of ZC-67nm particles (Fig. 1e and Supplementary Fig. 7). A thin Zr-rich oxide layer exists on the surface of ZC-67nm due to the slight oxidation in the ambient air (Supplementary Fig. 13).

Smelting-ZrCo generally requires activation before use, during which the alloy will pulverize and disproportionate (Supplementary Fig. 14). In this article, all smelting-ZrCo alloys refer to the activated samples unless otherwise stated. The activated ZC-177μm particles are polycrystalline and their crystallinity is significantly lower than that of chem-ZrCo (Supplementary Fig. 12b). According to the Rietveld refinement of XRD pattern, ZC-177μm contain 90.3% ZrCo, 8.1% ZrCo$_2$ and 1.6% ZrH$_2$ impurities, and its crystallite size is about 29.0 nm (Fig. 1c and Table 1), which is consistent with the HRTEM observation (Supplementary Fig. 14) and the literature reports[42]. In contrast, ZC-67nm nanoparticles do not require activation, and the particles maintain integrity, high crystallinity, and high purity after the initial cycle (Supplementary Fig. 15 and Table 1). This phenomenon could be attributed to the small size effect of nanoparticles, which is conducive to reducing the stress generated during hydrogenation/dehydrogenation.

## Anti-disproportionation performances of ZrCo

Figure 2a shows the disproportionation kinetics curves of ZrCo alloys at 500 °C and 0.7 MPa H$_2$ (and D$_2$ for ZC-67nm only). The disproportionation ratios of all smelting-ZrCo samples are over 80% within 200–400 min, much higher than those of chem-ZrCo. The disproportionation ratio of chem-ZrCo decreases with the particle size. ZC-67nm shows only 24% disproportionation over a long period of 1000 min. For deuterium absorption, the disproportionation ratio of

ZC-67nm is 29%. The XRD pattern shows that the ZC-67nm after disproportionation still contains ZrCoH$_3$ as the main phase (Supplementary Fig. 16a). According to the Rietveld refinement, the content of ZrCoH$_3$ is about 80.4% (Supplementary Table 2). In comparison, the disproportionated ZC-177μm is only composed of disproportionation phases (Supplementary Fig. 16c).

Generally, structural defect regions have high free energy, which facilitate the nucleation of new phases and the reduction of energy barrier of solid-state reaction. ZrCo particles often contain a large number of structural defects including grain boundary, phase boundary, surface, dislocation, inhomogeneities, etc. For single-crystal ZC-67nm with high purity and crystallinity, surface is the primary structural defects. As the particle size of ZrCo alloy increases, the crystallite size (obtained from Rietveld refinement) gradually decreases (Fig. 2b and Supplementary Table 3), implying a significant increase in the number of grain boundaries. Therefore, ZC-177μm particles contain high-density grain boundary defects, as well as a certain amount of phase boundaries, dislocations and inhomogeneities caused by low crystallinity and purity (Supplementary Fig. 14).

We used theoretical calculations to reveal the fundamental reasons for the easy occurrence of disproportionation reaction at structural defects, taking the main defects (i.e. surface and grain boundary) as examples. Using the ZrCoH$_3$ (002) crystal plane as a model, we calculated the formation energy ($E_f$) of inserting one H atom at the marked positions (Fig. 2c, d and Supplementary Figs. 17–19, and Tables 4, 5). When H was inserted near the surface (i.e. sites 2–4) and grain boundary (i.e. sites 3 and 4), the $E_f$ values at these positions are much lower than those at the internal positions of the grain. It suggests a more pronounced aggregation of H at structural defects, which is consistent with the recent reports on hydrogen trapping at the boundaries of Al alloys[43] and Zr alloys[44]. The increase in the number of hydrogen atoms will increase the nucleation probability of the disproportionated phases at these defect regions[45]. The energy changes of deuterium insertion in ZrCoD$_3$ present the same results as those of H in ZrCoH$_3$ (Supplementary Fig. 20 and Tables 6 and 7). Therefore, disproportionation reaction is more likely to occur at structural defects regions of ZrCo alloy, which can be regarded as the "zippers" to induce and promote the disproportionation reaction. For ZC-67nm, its structural defect is the boundary of nanoparticle and environment, i.e. the surface of the nanoparticle. We established a spherical model to calculate the grain boundary density ($\rho_b$) of ZC-67nm[46,47]. The $\rho_b$ of ZC-177μm was estimated by a Voronoi tessellation model (Supplementary Fig. 21)[48–50]. The $\rho_b$ of ZC-177μm is one order of magnitude higher than the $\rho_b$ of ZC-67nm (i.e. 0.586 nm$^{-1}$ vs. 0.088 nm$^{-1}$). Therefore, ZC-177μm is more prone to disproportionation reaction, consistent with the experimental results.

To experimentally prove the critical role of defects in promoting disproportionation reaction, we used bright- and dark-field TEM images to identify the positions of the ZrH$_2$ phase in ZC-67nm and ZC-177μm after 15 min and 1 h of disproportionation reactions. Among 20 samples of ZC-67nm/15 min, ZrH$_2$ phase is only observed on three samples. ZrH$_2$ distributes at the particle edge, and its area accounts for only 0.3% of the total particle area (Fig. 2e and Supplementary Fig. 22a). In contrast, ZrH$_2$ phase is observed on all ZC-177μm/15 min particles, and it mainly distributes inside the particles, accounting for 5.7% of the total area (Fig. 2f and Supplementary Figs. 22b and 23). After one hour of disproportionation, the area ratio of ZrH$_2$ in ZC-67nm/1 h and ZC-177μm/1 h increases to 4.2% (Fig. 2g and Supplementary Figs. 22c and 24) and 15.8% (Fig. 2h and Supplementary Figs. 22d and 25), respectively. The structural defects of single-crystal ZC-67nm distribute on the surface regions, while the defects of ZC-177μm mainly distribute inside the particles. The distribution positions of the disproportionation phase are consistent with those of structural defects, which also implies that the structural defects may act as zippers to initiate the disproportionation reaction. The defect mechanism was further

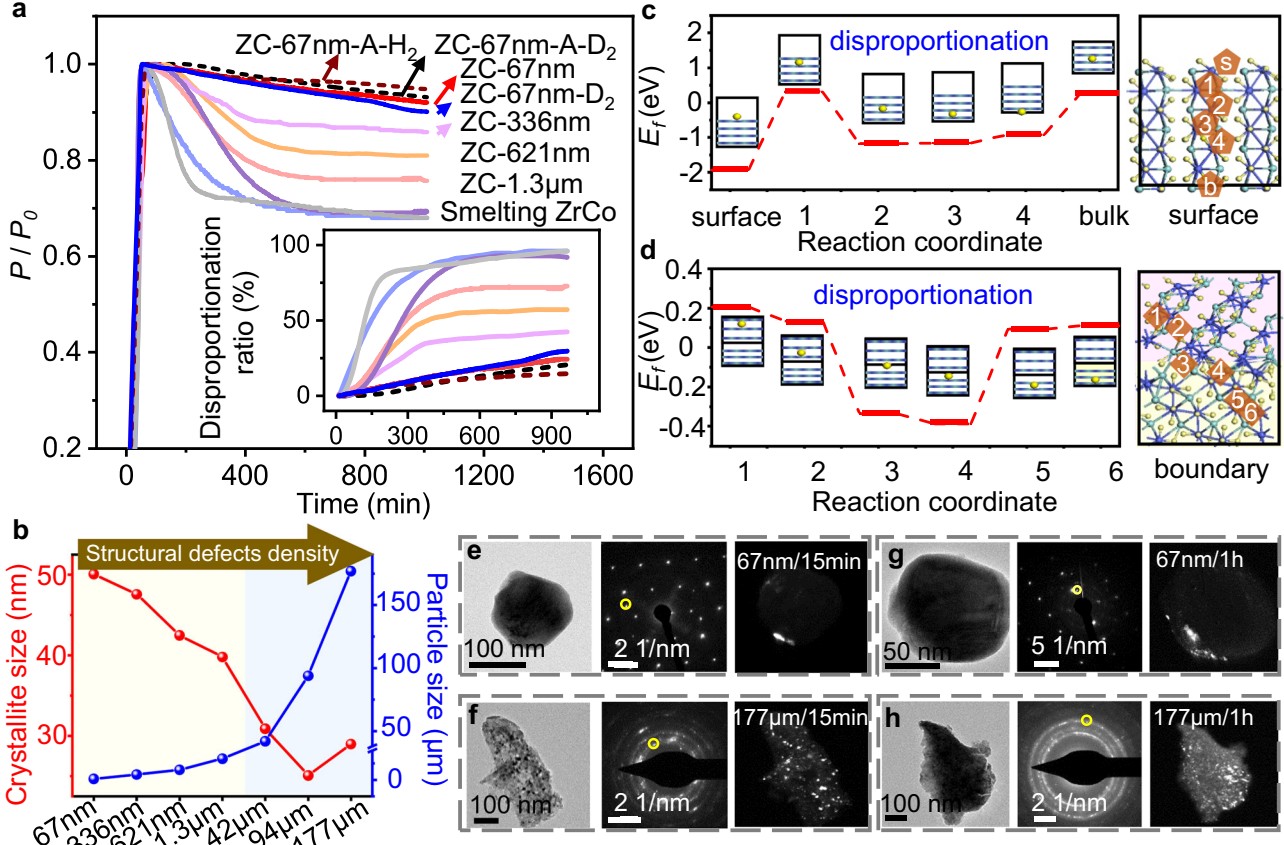

**Fig. 2 | Disproportionation of chem-ZrCo and smelting-ZrCo.**
**a** Disproportionation kinetics of ZrCo alloys at 500 °C and 0.7 MPa $H_2$ or $D_2$. ZC-67nm after annealing treatment is denoted as ZC-67nm-A. Inset: corresponding disproportionation ratio. **b** Variations of particle size and crystallite size of as-synthesized ZrCo alloys. Yellow region: chem-ZrCo, blue region: smelting-ZrCo. **c**, **d** Formation energy ($E_f$) of adding one H atom on the surface, subsurface and grain boundary of $ZrCoH_3$. The right side illustrates the positions of the added H in the $ZrCoH_3$ (002). The yellow, blue and green atoms correspond to H, Co and Zr atoms, respectively. Two different background colors indicate different orientations. **e**–**h** Bright-field TEM images of ZC-67nm and ZC-177μm after disproportionation reactions for 15 min and 1 h. The $ZrH_2$ phase in SAED is circled by yellow color, and highlighted in the dark-field TEM images.

proved by annealing experiment. As shown in Supplementary Figs. 26 and 27, the anti-disproportionation ability of ZrCo is improved after the annealing due to the decrease in the number of defects.

## $H_2$ and H-isotope storage performances
The storage performances of ZC-67nm for $H_2$ and $D_2$ are evaluated in terms of capacity, kinetics, thermodynamics, and cycling, and compared with other chem-ZrCo and smelting-ZrCo. The storage capacities of ZC-67nm for $H_2$ and $D_2$ at room temperature are 1.8 wt.% and 3.6 wt.%, respectively (Fig. 3a), close to the theoretical limitations. The molar storage capacities of $H_2$ and $D_2$ are the same. To uptake 90% of the maximum capacity, ZC-67nm only needs 8 s and 9 s for $H_2$ and $D_2$, respectively. In contrast, ZC-177μm requires 70 s to reach 90% of the maximum $H_2$ storage capacity. The isothermal hydrogenation curves at other temperatures (100, 200, and 300 °C) are shown in Supplementary Fig. 28. ZC-67nm exhibits the fastest kinetic performances at all temperatures (reaching 90% of the maximum capacity within 10 s). The superior kinetic performance of ZC-67nm can be attributed to the large surface area for hydrogen adsorption and short hydrogen diffusion path.

We conducted DFT calculations on the formation energy ($E_f$) of an interstitial H/D ($H_i/D_i$) in the subsurface layers of ZrCo (Fig. 3b, Supplementary Figs. 29–31, and Table 8). It can be seen that the H/D diffusion barrier derives from the penetration from surface to the first subsurface, and $D_i$ has a higher energy barrier than $H_i$. The $E_f$ of $H_i$ for ZrCo with (H1-H4 positions) and without (H5 position) defects (Zr/Co vacancy) was calculated to elucidate the effect of defects on the kinetics of ZrCo (Fig. 3c, Supplementary Figs. 32 and 33, and Table 9). It is found that the $H_i$ closest to the defect (i.e. H2 site) has the lowest $E_f$, indicating that defects readily trap H atoms and thus hinder the diffusion of H/H-isotope in the crystal. Therefore, the low defect density makes an important contribution to the excellent kinetic performance of ZC-67nm.

The thermodynamic properties of ZrCo alloys were evaluated using Pressure-Composition-Temperature (PCT) curves (Supplementary Fig. 34 and Table 10). The dehydrogenation enthalpy $\Delta H$ and entropy $\Delta S$ were calculated using the Van't Hoff formula. The plateau pressure was taken from the midpoint of the plateau region. As the particle size decreases from 177 μm to 67 nm, the $\Delta H$ value decreases from 99.7 to 95.7 kJ $mol^{-1}$, while the $\Delta S$ value decreases from 245.9 to 236.5 J $mol^{-1}$ $K^{-1}$ (Fig. 3d), indicating the improvement of the thermodynamic properties of the alloys. The $\Delta H$ and $\Delta S$ values for the deuterium desorption of ZC-67nm-D are 98.0 kJ $mol^{-1}$ and 241.5 J $mol^{-1}$ $K^{-1}$ respectively, which are higher than those of dehydrogenation of ZC-67nm-H[51,52] (Supplementary Figs. 35 and 34a). Temperature-programmed desorption (TPD) and differential scanning calorimetry (DSC) measurements show that the dehydrogenation temperature of ZrCo decreases with decreasing particle size, which can be attributed to the improved kinetic and thermodynamic properties (Fig. 3e, Supplementary Fig. 36 and Table 11). Low dehydrogenation temperature is of significance to resist disproportionation. Calculations indicate that the activation energy ($E_{de}$) of dehydrogenation decreases from 127.9 to

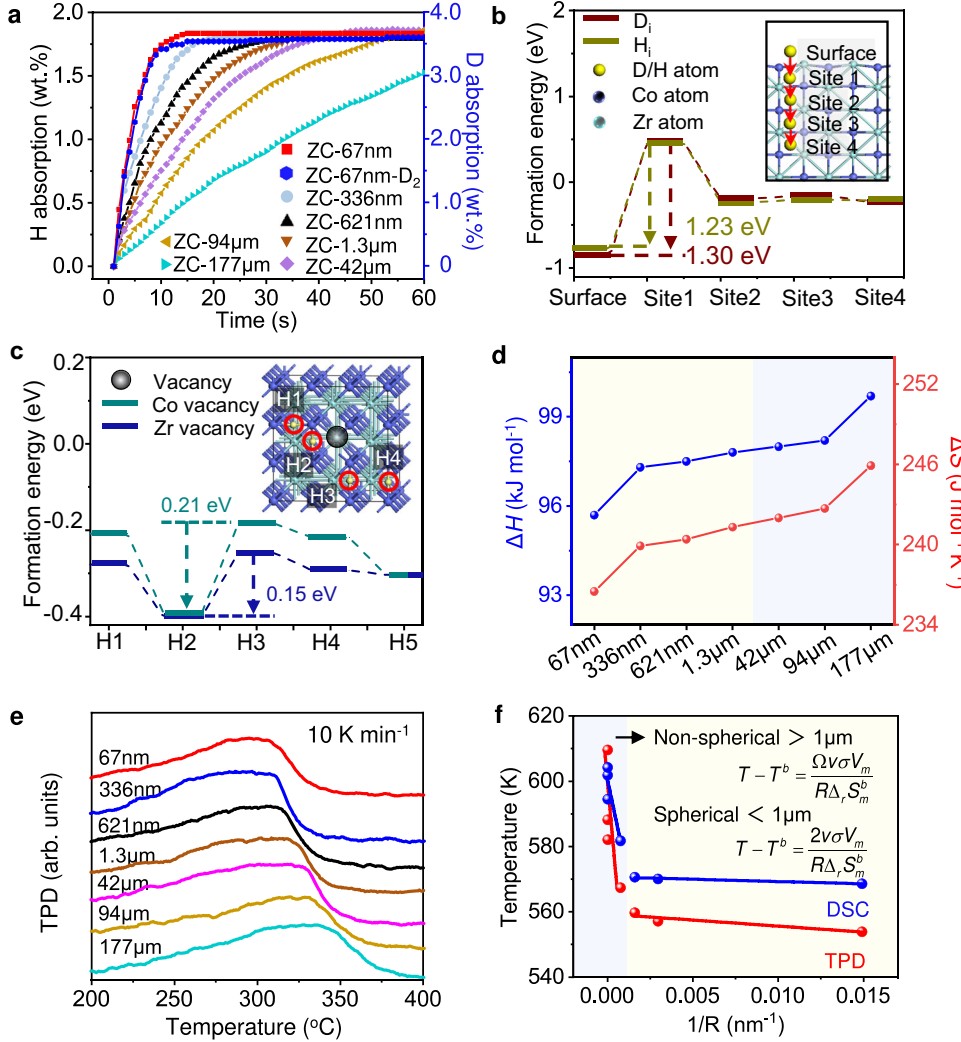

**Fig. 3 | H₂/D₂ storage performances of chem-ZrCo and smelting-ZrCo.**
**a** Isothermal hydrogen/deuterium absorption curves at room temperature.
**b** Formation energy ($E_f$) of single $H_i/D_i$ in the surface/subsurface of ZrCo (110) (red line indicates the diffusion path of $H_i/D_i$). The yellow, blue, and green atoms correspond to H, Co and Zr atoms, respectively. **c** $E_f$ of $H_i$ in ZrCo without (H5 site, shown in Supplementary Fig. 33) and with defects (H1-H4 sites, shown in the inset).

The blue and green atoms correspond to Co and Zr atoms, respectively. Green line and blue line correspond to the Co and Zr vacancies, respectively. **d** Variations of $\Delta H$ and $\Delta S$ with ZrCo particle size. Yellow region: chem-ZrCo, blue region: smelting-ZrCo. **e** TPD curves of ZrCo alloys. **f** The relationship of dehydrogenation temperatures ($T_{DSC}$, $T_{TPD}$) with reciprocal of particle radius ($1/R$). Yellow region: nanoscale ZrCo particles; Blue region: microscale ZrCo particles.

80.1 kJ mol⁻¹ as the particle size decreases from 177 μm to 67 nm (Supplementary Fig. 37). Moreover, it is interesting to find that the TPD peak temperature and DSC peak temperature have a linear relationship with the reciprocal of particle radius ($1/R$), i.e. $T = A/R + B$, as shown in Fig. 3f. The different slope A values for ZrCo alloys below and above 1 μm are due to the particle shape effect, as described in the Supplementary Methods. Using this formula, the dehydrogenation temperature of ZrCo of any size can be predicted.

At last, the cyclability of smelting-ZrCo and chem-ZrCo was evaluated as shown in Supplementary Fig. 38. As the particle size decreases from 177 μm to 67 nm, the hydrogen capacity retention of ZrCo boosts from 35% to 84% after 50 cycles. Similarly, the deuterium capacity retention of ZrCo increases from 33% to 83%. The superior cyclability of ZC-67nm is closely associated to the highly improved disproportionation resistance. ZC-67nm can retain more effective components than smelting-ZrCo during cycling. Based on the XRD patterns of hydrided ZC-67nm and ZC-177μm after 50 cycles (Supplementary Fig. 39 and Table 12), the ZrCoH₃ content in the hydrided ZC-67nm/50 cycles is as high as 90 wt.%, which is consistent with our experimental results. Additionally, the SEM and TEM images show that ZC-67nm

nanoparticles after 50 cycles can well retain their single-crystal structure, high crystallinity, and small size without pulverization and agglomeration (Supplementary Fig. 40). The high structural stability is also responsible for the excellent cyclability of ZC-67nm.

### Titanium doped single-crystalline nano-ZrCo

Based on the single-crystal nanoparticle strategy, doping Ti into ZrCo nanoparticles can further improve the anti-disproportionation and cycling properties of ZrCo alloy. We synthesized $Zr_{0.9}Ti_{0.1}Co$ nanoparticles using the same chemical method (Supplementary Fig. 41, Table 13), and compared its performances with ZC-67nm. The storage capacities of ZTC for H₂ and D₂ storage at room temperature are 1.7 wt.% and 3.4 wt.%, respectively (Fig. 4a). ZTC requires 11 and 12 s to absorb 90% of the maximum storage capacity of H₂ and D₂, respectively. Although the storage capacities and kinetics of ZTC slightly reduce compared with ZC-67nm, the anti-disproportionation ability of ZTC is significantly improved. Only quite slight disproportionation of 5% and 7% is observed for H₂ and D₂ storage after 1000 min, respectively (Fig. 4b, Supplementary Fig. 42a, and Table 14). Meanwhile, the cyclability of ZTC is also improved after 50 cycles, which is much better

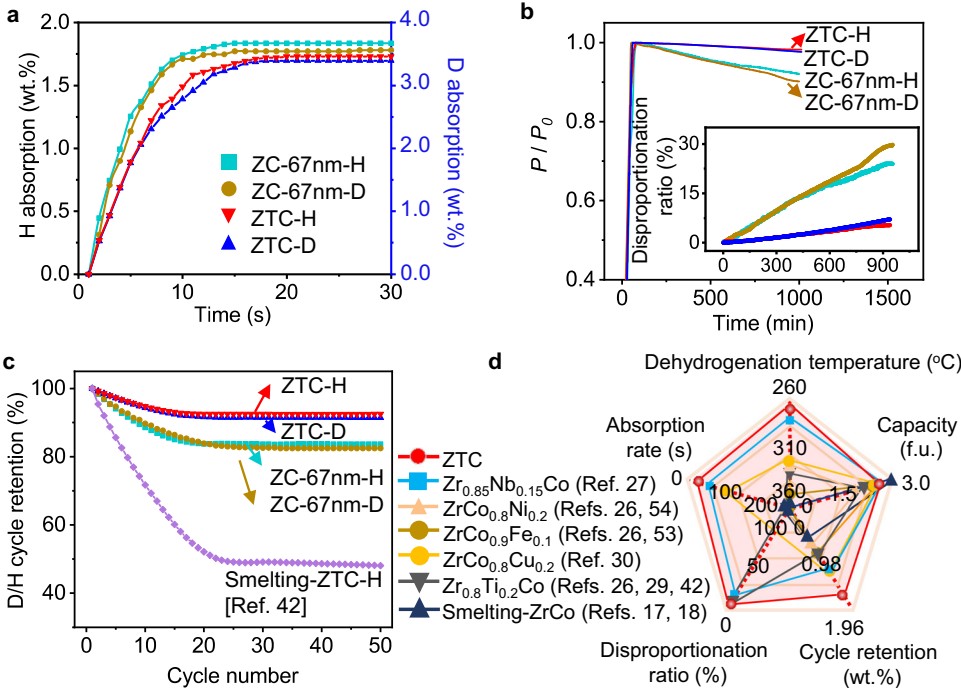

**Fig. 4 | Performance comparison of ZTC, ZC-67nm, smelting-ZTC, and comparison of comprehensive performances with other ZrCo alloys. a** Isothermal hydrogen/deuterium absorption curves at room temperature. **b** Disproportionation kinetics of ZrCo alloys at 500 °C, inset: corresponding disproportionation ratio. **c** The cyclability of as-synthesized ZrCo alloys. **d** A radar plot comparing the comprehensive hydrogen storage performances of ZrCo-based alloys.

than those of smelting-ZTC reported in literatures[42] (Fig. 4c, Supplementary Fig. 42b, and Table 14). TPD measurements show that the dehydrogenation peak temperature of ZTC is 271 °C, lower than that of ZC-67nm (281 °C; Supplementary Fig. 42c). And the dehydrogenation $\Delta H$ and $\Delta S$ values of ZTC are 79.7 kJ mol$^{-1}$ and 230.1 J mol$^{-1}$ K$^{-1}$, respectively (Supplementary Fig. 42d). Figure 4d and Supplementary Fig. 43 compare the key hydrogen storage properties of ZC-67nm and ZTC with other ZrCo-based alloys reported in literatures[30,53,54]. It can be seen that ZC-67nm and ZTC exhibit excellent comprehensive performances in terms of kinetics, anti-disproportionation, and cyclability. For conventional smelting-ZrCo alloy, the improvement of cycling performance through Ti doping is limited, with a cycling retention of only 48%. However, doping Ti into ZrCo nanoparticles can boost the cycling retention to 90%. This indicates the importance of single-crystal nanoparticle strategy in improving the comprehensive performance of ZrCo hydrogen storage alloys.

In addition, to avoid the potential risk of contaminating the storage and delivery systems of the ITER project in practical applications by nanoparticles, we further solved this problem by pressing ZC-67nm powders into a ring (millimeter size) (Supplementary Fig. 44). The ZrCo ring retained the excellent properties of ZC-67nm powder, including high H$_2$ storage capacity (~1.7 wt.%), fast kinetics (reaching 90% of the maximum capacity within 11 s at room temperature) and outstanding cyclability (77% capacity retention after 50 cycles). Even after 50 cycles, the ring maintains its complete shape.

**Nondestructive detection of hydrogen storage in ZrCo**

It is of practical significance to develop a non-destructive method to detect the H-isotope storage state and aging state of ZrCo. The changes of storage capacity, cycle and disproportionation can change the alloy phases and interfacial structure of ZrCo, which further affect the charge transport property and electric dipole polarization of the crystal. Therefore, we can use the sensitivity of microwaves to the charges and electric dipoles to nondestructively monitor the H/H-

isotope storage state of ZrCo (Fig. 5a). We measured the microwave transmission signal, and obtained the shielding effectiveness (*SE*) that generally increases with conductivity and polarization relaxation[55–57]. See Methods for sample preparation and testing.

As shown in Fig. 5b–d, the relative *SE* change ($\Delta SE/SE_0$) of each ZrCo alloy exhibits a linear relationship with hydrogen absorption ratio, cycle number, and disproportionation ratio. Taking ZC-67nm as an example, when the hydrogen storage capacity increases from 0 to 100%, the *SE* value gradually decreases from 6.1 to 1.9 dB (Fig. 5b and Supplementary Fig. 45a, b), because hydrogenation reduces the electrical conductivity of ZC-67nm (from 32.3×10$^4$ to 5.5×10$^4$ S m$^{-1}$, Supplementary Fig. 46a, b). During cycling test, the *SE* value of ZC-67nm continuously increases (Fig. 5c and Supplementary Fig. 45c, d) mainly due to the increase of heterogeneous interfaces where the accumulation of free charges in the alternating electromagnetic field produces strong space-charge polarization relaxation (Supplementary Fig. 46c, d). At last, as the disproportionation ratio of ZC-67nm increases from 0 to 25% within 16 h, the *SE* value increases from 1.9 to 6.2 dB (Fig. 5d and Supplementary Fig. 45e, f). This phenomenon associates with the increased conductivity and polarization relaxation due to the formation of disproportionation phases (Supplementary Fig. 46e, f). See Supplementary Fig. 46 for detailed explanation of the mechanism. ZC-1.3μm, ZC-42μm and ZC-177μm show a similar relationship (Supplementary Figs. 47, 50, 51), and this relationship has good repeatability (Supplementary Figs. 47–49). In addition, the validity of this microwave detection method was also verified on other representative hydrogen storage materials (Mg, LaNi$_5$, and TiFe) in Supplementary Fig. 52. These definite relationships enable us to estimate the H/H-isotope absorption state of any ZrCo material using microwaves.

In summary, owning to the low defect density, single-crystal ZrCo nanoparticles achieve high disproportionation resistance, which leads to the much better cyclability than conventional smelting ZrCo. Meanwhile, this material has high H$_2$/D$_2$ storage capacities (close to theoretical limits) and an order of magnitude higher kinetics than

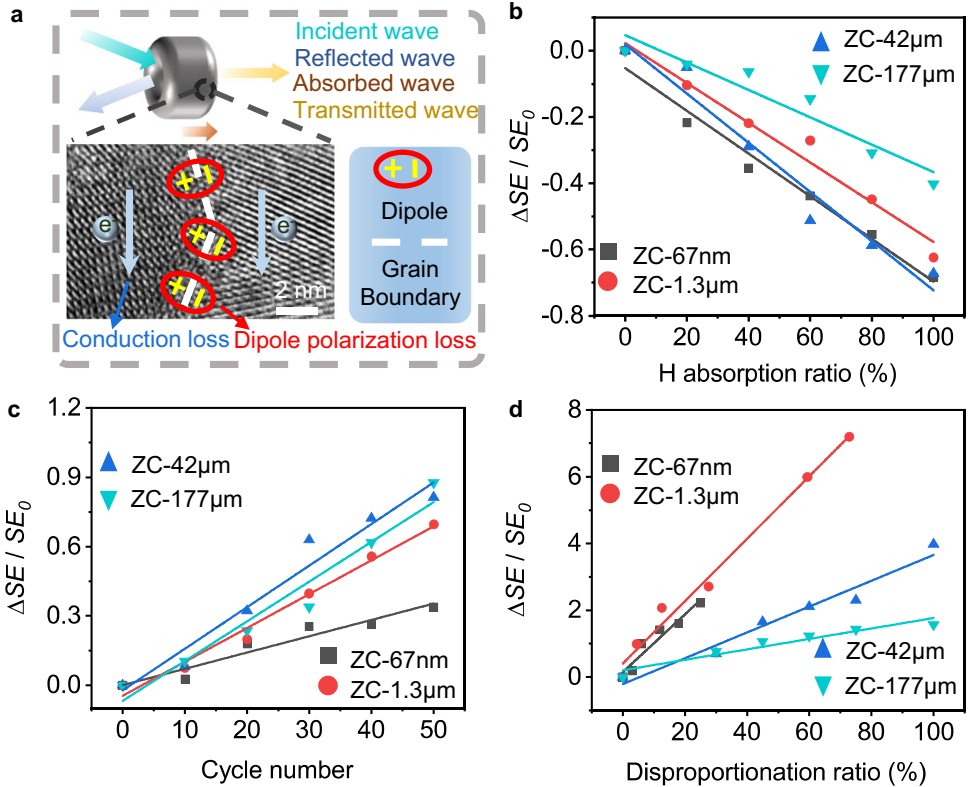

**Fig. 5 | Nondestructive detection of the hydrogen storage state of ZrCo by a microwave method. a** Schematic illustration of electromagnetic shielding mechanism in ZrCo. **b** Relative *SE* changes of ZrCo alloys with different hydrogen absorption ratios. **c** Relative *SE* changes of ZrCo alloys with different cycle numbers. **d** Relative *SE* changes of ZrCo alloys with different disproportionation ratios.

smelting-ZrCo. The much-improved performances will facilitate the application of safe and low-cost ZrCo alloy in controlled nuclear fusion in the future. Defect-induced disproportionation mechanism, nano-single crystal strategy and microwave detection method will also benefit other energy storage materials with the disproportionation problems.

## Methods

### Synthesis of chem-ZrCo

The synthesis of chem-ZrCo consists of three steps: synthesis of zirconium-cobalt hydroxides by co-precipitation, pyrolysis of hydroxides to oxides at moderate temperature, and magnesiothermic reduction of oxides. Specifically, zirconium-cobalt salt solutions ($Zr^{2+}$:$Co^{2+}$=1:1) with total metal cation concentrations of 0.05, 0.1, 0.3 mol $L^{-1}$ were prepared using $CoCl_2 \cdot 6H_2O$ and $ZrOCl_2 \cdot 8H_2O$ as raw materials. 4.5, 7.5, 16.5 ml of mixed alkali solutions of NaOH (0.35 mol $L^{-1}$) and $Na_2CO_3$ (0.15 mol $L^{-1}$) were added to 15 ml of the above three solutions, respectively, to completely precipitate the metal cations (PH = 10). After washing and centrifugation (at 1006 g) for 6 ~ 7 times, zirconium-cobalt hydroxides were obtained, and then re-dispersed in deionized water and frozen with liquid nitrogen. After freeze-drying, the fluffy blue-violet hydroxides were calcined in air at 450 °C for 3 h to form zirconium-cobalt oxides. Then, the oxides were reduced using magnesiothermic reduction method at 850 °C for 6 h under Ar atmosphere. The mass ratio of oxide powder to magnesium is 1:2. Finally, the ZrCo alloy was obtained by removing the generated magnesium oxide, residual magnesium, and other impurities using 0.1 M HCl, collecting and drying in vacuum at 45 °C for 6 h. The ZrCo nanoparticles (NPs) with an average particle size of 67 nm were prepared using a low metal salt concentration of 0.05 mol $L^{-1}$ combined with freeze-drying. As the metal salt concentration increased to 0.1 and 0.3 mol $L^{-1}$, the average particle size of ZrCo increased to 336 and 621 nm, respectively. With the same concentration of 0.3 mol $L^{-1}$, ZrCo

particles with an average size of 1.3 μm was obtained when the hydroxides were subjected to a general drying at 60 °C for 48 h in air. The above samples were denoted as ZC-67nm, ZC-336nm, ZC-621nm, and ZC-1.3μm, respectively. And the Ti substituted of single-crystal $Zr_{0.9}Ti_{0.1}Co$ is prepared by adding $TiCl_4$ during co-precipitation, and the subsequent preparation process is same as ZC-67nm. All ZrCo particles were stored in a glove box filled with Ar to prevent oxidation.

### Synthesis of smelting-ZrCo

Smelting-ZrCo was prepared via arc smelting method under Ar atmosphere. The purities of raw zirconium and cobalt metals are above 99.9%. The ingots were turned over and re-melted for 4 times to ensure homogeneity. After cooling to room temperature, the ingot was mechanically pulverized and sieved through 60, 200, and 400 mesh screens to obtain powders with average particle sizes of 42, 94, and 177 μm. The above samples are denoted as ZC-42μm, ZC-94μm, and ZC-177μm, respectively.

### Characterization

The crystal structure of the samples was characterized by a Rigaku D/max 2500·X-ray·diffractometer with Cu Kα radiation source ($\lambda$ = 0.15418 nm). A graphite monochromator was used to filter fluorescence. The morphology, microstructure, and elemental distribution of the samples were observed by field-emission scanning electron microscopy (FESEM, MERLIN VP Compact, Zeiss) and transmission electron microscopy (TEM, JEM-2100F, JEOL). Differential scanning calorimetry (DSC, Setsys Evounder, Setaram) measurements were performed using high purity flowing argon (99.999%, 50 mL min$^{-1}$). An X-ray photoelectron spectroscope (XPS, ESCALAB 250 Xi) with an Al Kα excitation source was applied to probe the surface composition. Temperature-programmed desorption curves were collected using an AutoChem II 2920 instrument with $N_2$ as the carrier gas. The electrical

conductivities of the samples were measured by a four-probe method using KDY$^{-1}$ digital meter (Guangzhou KunDe Co., Guangzhou, China) at room temperature.

## Hydrogen and H-isotope storage measurements

A Sieverts-type pressure-composition-temperature apparatus (PCT, General Research Institute for Nonferrous Metal, China) was used to measure the hydrogen and deuterium storage performances. Before the storage test, 200 mg of the sample was loaded into a sample vessel and degassed at 500 °C for 30 min under vacuum ($\leq 10^{-3}$ Pa). For the initial isothermal hydrogenation process, the sample was hydrogenated under 2 MPa H$_2$ (or D$_2$) at 200 °C for 1 h for full activation. Thereafter, the hydride phase was degassed at 500 °C for 30 min. Then the isothermal hydrogenation (at room temperature, 100, 200, and 300 °C) experiments were performed under 2 MPa. After each process, the hydrogenated sample was degassed again at 500 °C for 30 min under vacuum ($\leq 10^{-3}$ Pa) to ensure complete degassing. For cycle measurements, the activated sample was hydrogenated at 300 °C for 30 min with an initial pressure of 2 MPa and dehydriding followed time-temperature program to 380 °C with a heating rate of 5 °C min$^{-1}$. Then the reactor was air cooling for the next de-/hydriding cycle. For disproportionation measurement, the activated sample firstly absorbed hydrogen at 200 °C under an initial pressure of 2 MPa. The entire system was then cooled to room temperature and evacuated to slightly less than 0.7 MPa. As the reaction started, the hydrogenated sample was heated to 500 °C at a rate of 10 °C min$^{-1}$ and held for 1000 min. Consequently, the extent of disproportionation can be estimated using the pressure change of the system. For the annealing treatment, the sample was placed in the sample vessel of the PCT apparatus and then annealed in vacuum at 500 °C for 10 h.

## Measurement of electromagnetic interference shielding performance

The EMI SE of the samples was tested on the basis of the coaxial method using a vector network analyzer (VNA, N5234B PNA-L) in the frequency range of 8.2–12.4 GHz. ZrCo alloy has a high conductivity and thus can strongly reflect microwaves. In order to allow microwaves to enter the testing ring, we uniformly mixed ZrCo particles with paraffin (which has a negligible response to microwaves) to decrease the overall conductivity of the testing ring. The mass ratio of ZrCo alloy to paraffin was 2:1 (total weight 0.2 g). And then the mixture was pressed into a compact and flat ring at a pressure of 2 MPa. The ring has an outer diameter of 7.0 mm, an inner diameter of 3.04 mm, a thickness of ~2 mm and a density of 3.2 g cm$^{-3}$. Microwaves were perpendicularly incident on the ring. Before testing, a standard calibration process was performed.

## Data availability

All data are included in the article, the supplementary information, and the source data file.

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

## Acknowledgements

J.L.S. was supported by National Key Research and Development Program of China (2021YFB4000601), National Natural Science Foundation of China (22225903, 21975010, U21A20328), Beijing Natural Science Foundation (Z200012). R.Y. was supported by National Natural Science Foundation of China (51731002), National Key Research and Development Program of China (2021YBF3501304). X.F.Liu was supported by National Natural Science Foundation of China (52371171, 52222106, 51971008), Natural Science Foundation of Beijing Municipality (2212033).

## Author contributions

R.Y., J.L.S., X.F.Liu, Z.L. and Y.P. conceived the project. Z.L., Y.P., R.Z., X.F.Li, C.C., G.D., and P.Y. performed the experiments and testing. Z.L., G.H., and Y.Y. analyzed the data. S.L., J.X.S., and Z.L. conducted the DFT calculation. R.Z., H.Z., M.F., J.H., H.C., and X.S. validated the data. Z.L., X.F.Liu and J.L.S. co-wrote the manuscript. R.Y., X.F.Liu and J.L.S. supervised the project. All authors discussed the results and commented on the manuscript.

## Competing interests

The authors declare no competing interests.
