## [Peer Review File · Nature Communications]

REVIEWER COMMENTS

Reviewer #1 (Remarks to the Author):

Hydrogen-isotope storage materials are essential for the controlled nuclear fusion. This manuscript reveals a defect-derived disproportionation mechanism and reports a nano-single-crystal strategy to solve ZrCo's problems. However, there are some critical issues in the current manuscript:

1. Please carefully examine the mathematical derivation. There is a mathmatic error in fractional differential equation (18), which is the theoretical base of the relationship between dehydrogenation temperature and crystallite size.
2. Doping ZrCo with Nb can accelerate the activation kinetics for 10 times compared with pristine ZrCo. Specifically, the initial activation period was remarkably reduced from 87.79 h for ZrCo to 8.08 h for Zr_{0.8}Nb_{0.2}Co reported by Ref. 27. Moreover, it was reported that hafnium substitution ratio had little effect on hydrogen absorption rate from Ref. 25. Hence, the statements that substitution strategy significantly slows down the storage kinetics should be corrected.
3. The disproportionation kinetic curves of ZrCo alloys at 500 °C and 0.7 MPa H₂/D₂ all show pressure drop resulting from the hydriding disproportionation reaction ($2\text{ZrCo} + \text{H}_2 \rightarrow \text{ZrCo}_2 + \text{ZrH}_2$). Further, according to the thermodynamic properties of ZrCo alloys with different crystallite sizes (Table S9), the dissociation pressures at 500 °C can be calculated (0.2278 MPa - 0.5485 MPa for ZC-67nm - ZC-177nm samples, correspondingly). They are less than the disproportionation pressure (0.7 MPa), which indicates no existence of α phase at the beginning of insulation at 500 °C. Only dehydriding disproportionation reaction ($2\text{ZrCoH}_3 \rightarrow \text{ZrCo}_2 + \text{ZrH}_2 + 2\text{H}_2$) can occur under the disproportionation measurement condition which should cause a rise in system pressure. Please explain the contradiction between thermodynamic parameters and disproportionation kinetic curves.
4. ZrCo alloy with high absolute value of hydrogenation enthalpy change corresponds to a noticeable heat effect during hydrogenation process. Hence, the isothermal hydrogenation curves can be questionable for activation energy calculation.
5. (002) plane of ZrCoH₃ is neither the crystal plane with the highest diffraction intensity nor the preference crystal plane. Please explicit the reason for (002) plane as the simulation model.
6. In the H₂ and H-isotope storage performances section, "It can be seen that the H/D diffusion barrier derives from the penetration from surface to the first subsurface, and D_i has a higher energy barrier than H_i" which is not accordance with the Fig. 3b. The diffusion energy barrier of H_i and D_i is 1.23 eV and 1.10 eV, respectively.
7. Please carefully examine the legend of Fig. 5. The unit of entropy changes (ΔS) in Table S9 is wrong.
8. Please further polish the language of the entire manuscript. For example, in the Main section, there is a mistake: "ZrCo intermetallic compound is currently the only substitute for uranium with the advantages of safety, higher storage capacity and excellent ³He trapping ability (help obtain high purity fuel)."

Reviewer #2 (Remarks to the Author):

This article introduced a wet-chemical method for preparing nano-scale ZrCo alloy for hydrogen isotope storage. The alloys prepared in this method have a single-crystal structure and have a lower chance of disproportionation during hydrogenation, which gives them advantages in cycle performance. The author use the XRD, TEM, and theoretical calculation concluded that the single crystal structure which has the lowest defects is attributed to such high stability. Before the possible publication, some questions are as below:

1. On page 8, the author mentioned that “we used bright- and dark-field TEM images to identify the positions of the ZrH₂ phase”. It would be better to discuss why bright- and dark-field TEM images can distinguish ZrH₂ from ZrCo in the manuscript.
2. According to the author’s viewpoint, the alloy that has lower defects has a lower chance of disproportionation. Are there any quantitative methods to characterize the amounts of defects? The grain-grain boundary may bring defects to the crystal, but the grain-environment boundary is also boundaries that bring defects to the ZC-67nm. Furthermore, as the ZC-67nm is the smallest sample, it has the highest specific surface area, which means has the largest grain-environment boundary. So, the characterization of how these different boundaries affect the number of defects is important for the conclusion.
3. In Fig. S15, the spectrum for ZC-177 μ m has a very poor signal/noise ratio. Is such data suitable for XRD refinement? Also most refinement data supposed that all the contents are crystallized, what about the amorphous phase during the hydrogen storage?
4. On table S12, the XRD refinement used the Ti as the reference. Is that mean the doping Ti species are in independent Ti metal form rather than the Zr(Ti)Co?

Reviewer #3 (Remarks to the Author):

Manuscript on “Single-crystal ZrCo nanoparticle for advanced hydrogen and H-isotope Storage”

The authors present a very detailed investigation of hydrogen absorption properties of ZrCo alloys comparing conventional smelting preparation with a wet-chemistry method. Their measurements indicate a reduced disproportionation rate for smaller single crystals compared to polycrystals. Their main arguments are based on HRTEM micrographs on individual crystals, see Figures S20 to S23. The bright spots are assigned to Zr hydride. From this 2-dimensional transmission view the location of these spots is correlated to the grain boundaries or surface, however, the grains are not clearly visible, and therefore, this correlation could be wrong since other defects may be responsible, e.g. inhomogeneities or oxides. The disproportionation could occur by forming dislocation at the phase boundary between metal and hydride because of the lattice mismatch. In smaller crystals dislocation formation may be more difficult, however, this typically is for much smaller crystallite sizes. The observed surface oxide layers may have an additional influence on the smaller nanocrystals. The statistics are not too striking: "Among 20 samples of ZC-67nm/15min, ZrH₂ is only observed on 3 samples." Another point is the stability of the nanoparticles on cycling in a real, long-term application, usually agglomeration and crystal growth are occurring.

All in all, it is an interesting and thorough investigation, however, I am missing the real novelty or step beyond the current knowledge. The manuscript is certainly interesting for specialists, but less for the wider audience of Nature Communication. A materials science journal, e.g., Journal of Alloys and Compounds, would be more suitable.

Reviewer #4 (Remarks to the Author):

In this paper, the authors report for the first time a nanoscale ZrCo alloy that shows excellent comprehensive hydrogen/hydrogen isotope storage properties. It is also revealed that structural defects are the key cause of disproportionation reactions in this class of alloys. The investigation is systematic and the authors provide extensive data to support their conclusions. In addition, they innovatively propose a microwave method for nondestructive detection of the hydrogen storage state of ZrCo. I recommend publication after a minor revision. Here are some specific suggestions.

1. Generally, conventional ZrCo requires activation, during which disproportionation and pulverization occur, resulting in a loss of hydrogen storage capability. It is interesting that ZrCo nanoparticles do not require activation. In addition to the performance tests, the authors are suggested to further explain this phenomenon by combining with other characterizations, such as XRD and TEM.
2. Page 12: The authors associated the excellent cyclability of nano-ZrCo to its anti-disproportionation ability, and analyzed the phase abundance in nano-ZrCo after 50 cycles. However, besides disproportionation, are there other reasons that affect the cyclability, such as pulverization and

decreased crystallinity? Therefore, I suggest the authors use TEM to analyze the microstructure of the aged nano-ZrCo after the cycling.

3. The authors should provide Rwp factor of the Rietveld refinement to ensure the reliability of the results.

4. The authors compressed ZrCo nanoparticles into rings without sacrificing performance. However, when testing the electromagnetic interference shielding performance, the ring of ZrCo powders was mixed with paraffin. Why did the authors measure the ZrCo-paraffin ring instead of ZrCo ring? The reason should be provided in the method section.

5. In Fig. 5, the authors intend to use the background color to indicate the relative SE change trend of ZrCo alloys. However, due to the overlap of some background colors, it may cause misunderstandings. I suggest the authors remove the background color.

6. Based on the comprehensive hydrogen storage properties, ZrCo nanoparticles show attractive application prospects. Can this nanoscale material be prepared on a large scale to meet the needs of practical applications?

REVIEWER COMMENTS

Response to Reviewer #1:

Q1. Please carefully examine the mathematical derivation. There is a mathematic error in fractional differential equation (18), which is the theoretical base of the relationship between dehydrogenation temperature and crystallite size.

Response: Thank you very much for your comment. We speculate that the mathematic error in equation (18) mentioned by the reviewer is because we did not clearly explain “ σ is a function of T and r ”. Therefore, the partial derivative formula

$$\left(\frac{\partial\sigma}{\partial r}\right)_p = \left(\frac{\partial\sigma}{\partial T}\right)_p \left(\frac{\partial T}{\partial r}\right)_p + \left(\frac{\partial\sigma}{\partial r}\right)_p \left(\frac{\partial r}{\partial r}\right)_p$$

should be used. We double checked the equations and compared them with the relevant equations reported in the literatures (Ref. *Physica B* 2014, 454, 175). Equation (18) is correct. But in the original manuscript, the definition of some physical quantities is unclear, and we omitted several calculation steps, which may cause misunderstanding. Additionally, we revised the

order of magnitude of $\left(\frac{\partial\sigma}{\partial T}\right)_p$ according to literatures (*Int. Mater. Rev.* 1993, 38,

157). In this case, there is an approximate linear relationship between the decomposition temperature (T) and reciprocal of particle radius ($1/R$), *i.e.* $T=A/R+B$. Meanwhile, we considered the effect of particle shape on the $\Delta_r G_m^s$. We modified the calculation section in the revised Supplementary (Pages 6-9, highlighted).

In order to prove the correctness of the formula ($T=A/R+B$), we used this formula to fit the experimental data in our experiment (Fig. 3f) and the results reported in previous literatures (Refs. *Int. J. Hydrogen Energ.* 2014, 39, 13628; *Mater. Res. Bull.* 2008, 43, 1263; *J. Am. Chem. Soc.* 2008, 130, 6761). As shown below (Fig. R1), the Y-axis and X-axis are dehydrogenation temperature T and reciprocal of the particle radius ($1/R$), respectively. The fitting lines (dashed line) are in good agreement with the experimental data (scatters). Therefore, the proposed mathematical relationship between dehydrogenation temperature and ZrCo particle size should be correct and

universal.

Fig. R1. Plots of dehydrogenation temperature vs. reciprocal of the particle size: (a) Mg and C mixture milled for different hours (from Ref. *Int. J. Hydrogen Energ.* 2014, 39, 13628); (b) LiAlH₄ and Li₃AlH₆ (from Ref. *Mater. Res. Bull.* 2008, 43, 1263); (c) NaAlH₄ (from Ref. *J. Am. Chem. Soc.* 2008, 130, 6761).

Q2. Doping ZrCo with Nb can accelerate the activation kinetics for 10 times compared with pristine ZrCo. Specifically, the initial activation period was remarkably reduced from 87.79 h for ZrCo to 8.08 h for Zr_{0.8}Nb_{0.2}Co reported by Ref. 27. Moreover, it was reported that hafnium substitution ratio had little effect on hydrogen absorption rate from Ref. 25. Hence, the statements that substitution strategy significantly slows down the storage kinetics should be corrected.

Response: Thank you very much for pointing out this problem. We have revised the discussion about the conventional doping in the main text (Page 3, highlighted).

Q3. The disproportionation kinetic curves of ZrCo alloys at 500 °C and 0.7 MPa H₂/D₂ all show pressure drop resulting from the hydriding disproportionation reaction (2ZrCo+H₂→ZrCo₂+ZrH₂). Further, according to the thermodynamic properties of ZrCo alloys with different crystallite sizes (Table S9), the dissociation pressures at 500 °C can be calculated (0.2278 MPa-0.5485 MPa for ZC-67nm-ZC-177nm samples, correspondingly). They are less than the disproportionation pressure (0.7 MPa), which indicates no existence of α phase at the beginning of insulation at 500 °C. Only dehydriding disproportionation reaction (2ZrCoH₃→ZrCo₂+ZrH₂+2H₂) can occur under the disproportionation measurement condition which should cause a rise in system pressure. Please explain the contradiction between thermodynamic parameters and disproportionation kinetic curves.

Response: In original manuscript, we used a wrong Van't Hoff equation

$$\left(\ln \frac{P_{\text{eq}}}{P_0} = -\frac{\Delta H}{RT} + \frac{\Delta S}{R}\right)$$

for the dehydrogenation reactions. Therefore, we got an abnormally low entropy value (ΔS : 101.8-122.3 J mol⁻¹K⁻¹), which makes the calculated pressure lower than the disproportionation pressure of 0.7 MPa. In the revised manuscript, we corrected the Van't Hoff equation ($\ln P_{\text{eq}} = -\frac{\Delta H}{RT} + \frac{\Delta S}{R}$) according to literature reports (Ref. *J. Mater. Chem. A* 2020, 8, 9322; *Int. J. Hydrogen Energ.* 2019, 44, 28242; *J. Alloy Comp.* 2020, 848, 156618). The recalculated ΔS values are shown in Supplementary Table S10 (also given below). Based on these values, the calculated dissociation pressures at 500 °C should be 22730.5-54069.9 MPa, much higher than the disproportionation pressure of 0.7 MPa. In this case, the hydriding disproportionation reaction occurred, leading to a pressure drop, which is consistent with our experimental results.

Table R1. A summary of enthalpy (ΔH) and entropy (ΔS) values of ZrCo alloys.

Sample	ΔH (kJ mol ⁻¹)	ΔS (J mol ⁻¹ K ⁻¹)
ZC-67nm	73.4 ± 0.4	197.5 ± 0.6
ZC-336nm	74.3 ± 1.6	199.4 ± 2.7
ZC-621nm	75.5 ± 1.4	201.9 ± 2.3
ZC-1.3μm	77.0 ± 3.2	204.9 ± 5.5
ZC-42μm	80.9 ± 2.8	212.5 ± 4.8

ZC-94 μm	81.8 ± 2.1	214.4 ± 3.5
ZC-177 μm	83.6 ± 2.8	217.9 ± 4.9

Q4. ZrCo alloy with high absolute value of hydrogenation enthalpy change corresponds to a noticeable heat effect during hydrogenation process. Hence, the isothermal hydrogenation curves can be questionable for activation energy calculation.

Response: Thank you for your suggestion. We did not calculate the hydrogenation activation energy based on the isothermal hydrogenation curves in our work. The isothermal hydrogenation curves in Fig. 4a and Supplementary Fig. S28 only indicate that the hydrogen absorption rate of the activated ZrCo alloys increases with the decrease of particle size. In the manuscript, we calculated the dehydrogenation activation energy (E_{de}) based on the DSC curves with different heating rates in Supplementary Fig. S37.

Q5. (002) plane of ZrCoH₃ is neither the crystal plane with the highest diffraction intensity nor the preference crystal plane. Please explicit the reason for (002) plane as the simulation model.

Response: Thank you for your comment. According to the standard PDF card (JCPDS#47-1411), the (111) plane of ZrCo is the crystal plane with the highest diffraction intensity, and the (002) plane is the crystal plane with the secondary highest intensity. In our XRD pattern of hydrided ZC-177 μm (Fig. R2a), the diffraction intensity of the (002) plane of ZrCoH₃ is only slightly lower than that of the (111) plane. Meanwhile, we can clearly observe the (002) crystal planes of ZrCoH₃ on the surface and inside the hydrided ZC-177 μm particle (Fig. R2b). Therefore, (002) crystal plane can be used as a representative crystal plane for ZrCoH₃ phase. Moreover, as for the calculation, the (002) plane of ZrCoH₃ has a lower miller indices and better symmetry compared to the (111) plane, which makes it easier to study the location of disproportionation reaction.

Fig. R2. XRD pattern and HRTEM image of hydride ZC-177µm.

Q6. In the H₂ and H-isotope storage performances section, “It can be seen that the H/D diffusion barrier derives from the penetration from surface to the first subsurface, and D_i has a higher energy barrier than H_i” which is not accordance with the Fig. 3b. The diffusion energy barrier of H_i and D_i is 1.23 eV and 1.10 eV, respectively.

Response: We apologize for this mistake. The descriptive text in the original manuscript is correct. But we used the incorrect calculation results of D_i (without phonon correction) in the previous Fig. 3b. In the revised manuscript, we redraw the curve of D_i with phonon correction. The new figure is also shown in Fig. R3. The corrected diffusion energy barrier of D_i is 1.30 eV.

Fig. R3. Formation energy (E_f) of single H_i/D_i in the surface/subsurface of ZrCo (110) (red line indicates the diffusion path of H_i/D_i).

Q7. Please carefully examine the legend of Fig. 5. The unit of entropy changes (ΔS) in Table S9 is wrong.

Response: We have revised the legend of Fig. 5 and the unit of entropy changes (ΔS) in Supplementary Table S10 (*i.e.* previous Supplementary Table S9) in the revised manuscript.

Q8. Please further polish the language of the entire manuscript. For example, in the Main section, there is a mistake: “ZrCo intermetallic compound is currently the only substitute for uranium with the advantages of safety, higher storage capacity and excellent ^3He trapping ability (help obtain high purity fuel).”

Response: Thank you for your suggestion. We have carefully polished the language of the manuscript.

Response to Reviewer #2:

Q1. On page 8, the author mentioned that “we used bright- and dark-field TEM images to identify the positions of the ZrH₂ phase”. It would be better to discuss why bright- and dark-field TEM images can distinguish ZrH₂ from ZrCo in the manuscript.

Response: Thank you for your suggestion. Generally, bright-field TEM images were obtained using the transmitted electron beams that pass through the objective aperture, while dark-field TEM images were obtained using the selected diffraction beams that pass through the objective aperture. First, we used objective aperture to select the electron beams passing through ZrCo particle, and obtained bright-field TEM image and corresponding SAED pattern. These results show the microstructure of ZrCo particle and the diffraction spots from ZrH₂ and ZrCo phases. After identifying the diffraction spots of ZrH₂ phase in the SAED pattern, we moved the objective aperture to only allow the diffraction beams of ZrH₂ phase to pass through. Therefore, the ZrH₂ phase presents bright spots due to its diffraction contrast in the dark-field TEM images, which can clearly show its distribution in the particle. This technique has been widely used to characterize the distribution of impurity phase, precipitate and disproportionate phase in the matrix. For example: *Science* 2017, 357, 1029; *Acta Mater.* 2019, 165, 228; *Mater. Today Energy* 2020, 18, 100554; *Adv. Eng. Mater.* 2020, 22, 1901079). Relevant discuss was provided in the revised Supplementary (Page 2-3, highlighted).

Q2. According to the author’s viewpoint, the alloy that has lower defects has a lower chance of disproportionation. Are there any quantitative methods to characterize the amounts of defects? The grain-grain boundary may bring defects to the crystal, but the grain-environment boundary is also boundaries that bring defects to the ZC-67nm. Furthermore, as the ZC-67nm is the smallest sample, it has the highest specific surface area, which means has the largest grain-environment boundary. So, the characterization of how these different boundaries affect the number of defects is important for the conclusion.

Response: According to your valuable suggestion, we established 3D models to

quantitatively analyze the grain-grain boundary density (ρ_{GG}) in ZC-177 μm and the grain-environment boundary density (ρ_{GE} , *i.e.* the specific surface area) in ZC-67nm.

(1) Grain-environment boundary density (ρ_{GE}) in ZC-67nm.

As shown in TEM images, ZC-67nm exhibits quasi-spherical particles. According to the literatures (Ref. *ACS Appl. Mater. Interfaces* 2017, 9, 2606; *Polymer* 2005, 46, 553), we established a spherical model to simulate ZC-67nm. Therefore, the grain-environment boundary density (ρ_{GE}) can be calculated as:

$$\rho_{GE} = \frac{S_{\text{sph}}}{V_{\text{sph}}} = \frac{\pi r^2}{\frac{4}{3}\pi r^3} = \frac{3}{4r}$$

where r , S_{sph} and V_{sph} are the radius, surface area and volume of the sphere. Here, $r = 34$ nm, and the ρ_{GE} of ZC-67nm should be **0.022 nm⁻¹**.

(2) Grain-grain boundary density (ρ_{GG}) in ZC-177 μm .

According to the literature reports (Ref. *J. Am. Ceram. Soc.* 2020, 103, 5900; *Comp. Mater. Sci.* 2022, 205, 111212; *Int. J. Fatigue* 2022, 156, 106680), we established a Voronoi tessellation model to simulate the polycrystalline structure of ZC-177 μm , as shown in Fig. R4. The Voronoi tessellation diagram was generated by Python, and the model containing the Voronoi tessellation diagram was built using ABAQUS software. The distribution of crystallite size was controlled by a coefficient that affects the distance between two seeds. We established a model containing 100 grains with an average size of 29 nm (based on the Rietveld refinement results of XRD pattern).

Fig. R4. (a, b) The 3D Voronoi tessellation model. (c) Side view and (d) the crystallite size distribution of ZC-177 μm .

The 3D ρ_{GG} of ZC-177 μm can be calculated:

$$\rho_{GG} = \frac{\sum_{100} S_{\text{voro}}}{\sum_{100} V_{\text{voro}}}$$

where S_{voro} and V_{voro} is the surface area and volume of each voro-module, both of which can be obtained statistically. Here, the total S_{voro} and V_{voro} are $\sim 325240.43 \text{ nm}^2$ and $\sim 555,209.56 \text{ nm}^3$, respectively. The ρ_{GG} of ZC-177 μm is about **0.586 nm^{-1}** .

Based on the above quantitative calculations, we can conclude that the ρ_{GG} value of ZC-177 μm is one order of magnitude higher than that of ZC-67nm (*i.e.* 0.586 nm^{-1} vs. 0.022 nm^{-1}). Therefore, disproportionation reaction is more likely to occur in ZC-177 μm , which is consistent with our experimental results. Relevant discuss was provided in the revised main text (Page 10, highlighted) and Supplementary Fig. S21 (Pages 33-34).

Q3. In Fig. S15, the spectrum for ZC-177 μm has a very poor signal/noise ratio. Is such data suitable for XRD refinement? Also most refinement data supposed that all the

contents are crystallized, what about the amorphous phase during the hydrogen storage?

Response: We agree with the reviewer. The previous XRD pattern of ZC-177 μm (after disproportionation) has a poor signal/noise ratio, which is not suitable for XRD refinement. Moreover, amorphous phase formed in this sample during the hydrogen storage. Therefore, XRD refinement is not suitable for obtaining the phase abundance of this sample. In previous Supplementary Fig. S15, we only attempted to demonstrate that ZC-177 μm after disproportionation did not contain ZrCoH_3 phase. Therefore, it is not necessary to conduct the XRD refinement.

In the revised manuscript, we provided a new XRD pattern of ZC-177 μm with higher signal/noise ratio (Supplementary Fig. S16c, also shown below), which was measured using a high-intensity XRD equipment (Rigaku Smart Lab SE, 9 kW, 40 A). The XRD pattern can clearly indicate that ZC-177 μm is completely disproportionated and does not contain ZrCoH_3 phase.

Fig. R5. XRD pattern of ZC-177 μm after disproportionation.

Q4. On table S12, the XRD refinement used the Ti as the reference. Is that mean the doping Ti species are in independent Ti metal form rather than the $\text{Zr}(\text{Ti})\text{Co}$?

Response: The form of Ti species depends on the doping content in ZrCo. In fact, we prepared Ti-doped ZrCo nanoparticles with different Ti contents (from 10 at.% to 30 at.%). The XRD patterns and corresponding Rietveld refinement results indicate that metallic Ti phase appears in the $\text{Zr}_{0.8}\text{Ti}_{0.2}\text{Co}$ and $\text{Zr}_{0.7}\text{Ti}_{0.3}\text{Co}$ samples due to the high doping amount. But the Ti species in $\text{Zr}_{0.9}\text{Ti}_{0.1}\text{Co}$ only exists in the form of $\text{Zr}(\text{Ti})\text{Co}$.

To avoid this misunderstanding, we deleted the Ti information (Ti content is 0 at% for $Zr_{0.9}Ti_{0.1}Co$) from the phase abundance list (*i.e.* new Supplementary Table S13).

Response to Reviewer #3:

Q1. The authors present a very detailed investigation of hydrogen absorption properties of ZrCo alloys comparing conventional smelting preparation with a wet-chemistry method. Their measurements indicate a reduced disproportionation rate for smaller single crystals compared to polycrystals. Their main arguments are based on HRTEM micrographs on individual crystals, see Figs. S20 to S23. The bright spots are assigned to Zr hydride. From this 2-dimensional transmission view the location of these spots is correlated to the grain boundaries or surface, however, the grains are not clearly visible, and therefore, this correlation could be wrong since other defects may be responsible, e.g. inhomogeneities or oxides. The disproportionation could occur by forming dislocation at the phase boundary between metal and hydride because of the lattice mismatch. In smaller crystals dislocation formation may be more difficult, however, this typically is for much smaller crystallite sizes. The observed surface oxide layers may have an additional influence on the smaller nanocrystals. The statistics are not too striking: “Among 20 samples of ZC-67nm/15min, ZrH₂ is only observed on 3 samples.”

Response: We appreciate reviewer’s constructive comments. We agree with the reviewer’s viewpoints. Disproportionation readily occurs at structure defects including dislocation, inhomogeneities, grain boundary, phase boundary (between metal and hydrides/oxides/impurities) and surface. In original manuscript, we only emphasized the grain boundary defects. After revision, we confirmed that all types of defects contribute to disproportionation. The activated smelting-ZrCo particles contain all above structural defects as shown in Fig. R6. Since most of these structural defects (such as dislocation, inhomogeneities, grain boundary, phase boundary) exist inside the smelting-ZrCo particles, the observed bright spots (assigned to disproportionated phase ZrH₂) mainly appear inside the particles. In contrast, ZC-67nm nanoparticles are single crystals with high crystallinity and high purity. They mainly contain two types of defects, *i.e.* surface and phase boundary on the surface (between ZrCo and surface oxide layer). Therefore, the bright spots appear on the particle surface. In a word, the distribution positions of ZrH₂ (on particle surface or inside particle) is consistent with

the positions of structural defects. Detailed discussion has been revised on pages 8-10 (highlighted).

Fig. R6. (a) TEM and SAED images of ZC-177μm after activation; (b-d) HRTEM images of the indicated areas I, II and III. Red circles indicate inhomogeneities (amorphous phases); blue circles indicate phase boundaries (between metal and oxides); yellow circles indicate grain boundaries.

Q2. Another point is the stability of the nanoparticles on cycling in a real, long-term application, usually agglomeration and crystal growth are occurring.

Response: The atmosphere and temperature used for cycling testing are similar to the practical application conditions. And these testing conditions are also widely used in the cycling performance evaluation of ZrCo alloys (Ref. *Chem Eng. J.* 2023, 455, 140571, *J. Alloys Compd.* 2019, 784, 1062). The measurement results indicate that ZrCo nanoparticles have excellent cycling performance.

In the revised manuscript, we provided the SEM and TEM images of ZC-67nm after 50 cycles in Supplementary Fig. S40 (also shown in Fig. R7). These nanoparticles are still monodisperse with an average size of ~67 nm. No obvious agglomeration and crystal growth were observed. Relevant discuss was provided in the revised main text (Page 14, highlighted).

Fig. R7. SEM image of ZC-67nm after 50 cycles and the distribution of particle size.

Q3. All in all, it is an interesting and thorough investigation, however, I am missing the real novelty or step beyond the current knowledge. The manuscript is certainly interesting for specialists, but less for the wider audience of Nature Communication. A materials science journal, e.g., Journal of Alloys and Compounds, would be more suitable.

Response: We are pleased to highlight the innovation and scientific significance of this work.

The existing ZrCo alloys generally have two problems in practical application: (1) rapid performance degradation during cycling due to disproportionation, (2) requiring a long-term activation process (several hours).

The 8e site of ZrCo crystal is conventionally regarded as the origin of disproportionation. In our work, we investigated the disproportionation mechanism from a new perspective, and revealed for the first time that structural defects can initiate the disproportionation reactions in ZrCo alloys. Subsequently, we proposed a nano-single-crystal strategy to suppress disproportionation by significantly reducing defect density. This strategy not only significantly improved the anti-disproportionation ability and cyclicality of ZrCo, but also boosted the hydrogenation/dehydrogenation kinetics by an order of magnitude. Compared with conventional ZrCo alloys that require several hours of activation, our ZrCo nanoparticles does not require any activation process.

The proposed anti-disproportionation strategy is also instructive for other energy storage materials with disproportionation problems, such as electrode materials of most

batteries. Meanwhile, the synthetic method is innovative for ZrCo materials, which are generally prepared by smelting method in previous reports. Our work provides a reference for the preparation of multicomponent nano-alloys that are difficult to synthesize by conventional chemical reduction methods.

Moreover, we established a mathematical relationship between dehydrogenation temperature and ZrCo particle size, which has never been established before and is meaningful for predicting the dehydrogenation temperature of ZrCo of any size. Last but not least, we are the first to develop a microwave detection method to non-destructively monitor the working state of ZrCo.

Response to Reviewer #4:

Q1. Generally, conventional ZrCo requires activation, during which disproportionation and pulverization occur, resulting in a loss of hydrogen storage capability. It is interesting that ZrCo nanoparticles do not require activation. In addition to the performance tests, the authors are suggested to further explain this phenomenon by combining with other characterizations, such as XRD and TEM.

Response: Thank you very much for your suggestion. Smelting-ZrCo needs activation to obtain high reaction activity because of its large particle size and the low specific surface area. During the activation process, a large number of cracks appear in the smelting-ZrCo particles, significantly increasing the specific surface area and producing fresh active surface. The crystallite size decreases sharply due to the expansion/contraction of the lattice during activation. In contrast, the ZrCo nanoparticles have small size and high specific surface, which is beneficial for improving activity. We used XRD, SEM and TEM to characterize the ZrCo nanoparticles (ZC-67nm) after hydrogenation/dehydrogenation cycle (Supplementary Fig. S15, also shown in Fig. R8). The Rietveld refinement of XRD pattern shows the phase composition of ZC-67nm after activation remains almost unchanged. The SEM and TEM images show that ZC-67nm nanoparticles remain intact in morphology and high crystallinity through the activation. There is no obvious change in ZC-67nm before and after activation. Therefore, ZrCo nanoparticles do not require activation. Relevant discuss was provided in the revised main text (Pages 6-7, highlighted).

Fig. R8. (a) XRD pattern and Rietveld refinement result; (b) SEM, (c) TEM, SAED and (d) HRTEM images of ZC-67nm after the initial hydrogenation/dehydrogenation cycle.

Q2. Page 12: The authors associated the excellent cyclability of nano-ZrCo to its anti-disproportionation ability, and analyzed the phase abundance in nano-ZrCo after 50 cycles. However, besides disproportionation, are there other reasons that affect the cyclability, such as pulverization and decreased crystallinity? Therefore, I suggest the authors use TEM to analyze the microstructure of the aged nano-ZrCo after the cycling.

Response: According to the suggestion, we observed the microstructure of the aged nano-ZrCo using SEM and TEM. (Supplementary Fig. S40). It is clear that the ZC-67nm nanoparticles after 50 cycles retain their single-crystal structure, high crystallinity and small size without pulverization and agglomeration. Therefore, high structural stability is also an important reason for the good cycling performance of ZC-67nm. Relevant discuss was provided in the revised main text (Page 14, highlighted).

Fig. R9. (a) SEM image of ZC-67nm after 50 cycles and the distribution of particle size. (b) TEM, SAED and HRTEM of ZC-67nm after 50 cycles.

Q3. The authors should provide Rwp factor of the Rietveld refinement to ensure the reliability of the results.

Response: Thank you for the suggestion. We provided the Rwp factors of all Rietveld refinement in the revised manuscript.

Q4. The authors compressed ZrCo nanoparticles into rings without sacrificing performance. However, when testing the electromagnetic interference shielding performance, the ring of ZrCo powders was mixed with paraffin. Why did the authors measure the ZrCo-paraffin ring instead of ZrCo ring? The reason should be provided in the method section.

Response: We prepared two types of ZrCo rings with and without paraffin for different purposes. The pure ZrCo rings were used to measure the hydrogen storage performance. Experiments show that pure ZrCo ring not only retains the excellent hydrogen storage performance of powders, but also prevent the contamination of ITER systems by nanoparticles in practical applications.

However, the pure ZrCo ring has a high conductivity, which will strongly reflect

electromagnetic waves (EMWs). In order to allow EMWs to enter the testing ring, we mixed ZrCo particles with paraffin (which has a negligible response to EMWs) to decrease the overall conductivity of the testing ring. This method has been widely used in the measurements of electromagnetic wave absorption materials. For example, Ref. *Matter* 2021, 4, 1735, *ACS Nano* 2022, 16, 14490, *Adv. Funct. Mater.* 2019, 29, 1807624. Relevant discuss was provided on pages 21-22 in the revised main text (highlighted).

Q5. In Fig. 5, the authors intend to use the background color to indicate the relative SE change trend of ZrCo alloys. However, due to the overlap of some background colors, it may cause misunderstandings. I suggest the authors remove the background color.

Response: Thank you very much for this suggestion. We removed the background color in Fig. 5.

Q6. Based on the comprehensive hydrogen storage properties, ZrCo nanoparticles show attractive application prospects. Can this nanoscale material be prepared on a large scale to meet the needs of practical applications?

Response: Yes. They can be prepared on a large scale. The synthesis of nano-ZrCo involves two techniques: co-precipitation and magnesiothermic reduction. Both techniques can be easily scaled up (Ref. *Small:Methods* 2018, 2, 1800062). Therefore, we believe that nano-ZrCo can be prepared on a large scale to meet the needs of practical applications, usually hundreds of grams to several kilograms of ZrCo (Ref. *Fusion Sci. Technol.* 2009, 56, 856).

REVIEWER COMMENTS

Reviewer #1 (Remarks to the Author):

This manuscript reports the effects and mechanism of Ti, Cu, Y, La, Ce, Pr, Nd and Sm substitutions on the anti-disproportionation performance of ZrCo alloy by combining experiments and first-principles calculations. The authors have revised the manuscript and the problems are well solved, whereas there is still a critical issue in the revised manuscript. It should be pointed out that the only difference of P_{eq} between different Van't Hoff equation forms ($\ln(P_{eq}/P_0) = -\Delta H/(RT) + \Delta S/R$, $\ln P_{eq} = -\Delta H/(RT) + \Delta S/R$) is the unit of P_{eq} , which means that the unit of P_{eq} is Pascal for $\ln P_{eq} = -\Delta H/(RT) + \Delta S/R$, while bar for $\ln(P_{eq}/P_0) = -\Delta H/(RT) + \Delta S/R$. The dissociation pressures at 500 °C calculated to be 22730.5-54069.9 MPa in the ZrCo-H system are unreasonably high. The contradiction between thermodynamic parameters and disproportionation kinetic curves is not yet precisely understood. Please carefully double check the thermodynamic parameters for dissociation process.

Reviewer #2 (Remarks to the Author):

The revision is acceptable.

Reviewer #3 (Remarks to the Author):

Revised manuscript on "Single-crystal ZrCo nanoparticle for advanced hydrogen and H-isotope Storage"

The authors improved the manuscript and commented on most of the questions raised. If the details on the TEM investigations and XRD refinements are satisfactorily answered, I shall leave them to the expert reviewers in these fields. Reading the manuscript and the SI carefully, I found some open questions and inconsistencies.

XRD: The XRD patterns are shown as a function of 2 Theta, however, neither in the manuscript nor in the SI, the wavelength of the radiation is given. What X-ray source was used? Was there any monochromator applied, e.g. to avoid fluorescence of Co? On page 6 "The crystallite size of ZC-67nm was calculated to be 50.1 nm," how was the average crystallite size calculated, which diffraction peaks have been used?

Response to reviewer #2, Q2: “grain-environment boundary density (ρ_{GE} , i.e. the specific surface area)” the authors connect the grain boundary density to the specific surface area, however, firstly the two quantities have different units. Specific surface area is given in square meters per gram, therefore, eq (29) in the SI is wrong. In the response to Q2, the authors exchanged “specific surface area” for “grain-environment boundary density”, however, SI page 33 is still different. What is correct now? What is the definition of a “grain-environment boundary density”? Did the authors measure the specific surface area by gas adsorption techniques?

Reviewer #4 (Remarks to the Author):

The authors have given reasonable answers and explanations to my questions, and the manuscript has been revised accordingly and properly. I think this revised manuscript should be published as is.

Response to Reviewer #1:

Q. The authors have revised the manuscript and the problems are well solved, whereas there is still a critical issue in the revised manuscript. It should be pointed out that the only difference of P_{eq} between different Van't Hoff equation forms ($\ln(P_{eq}/P_0) = -\Delta H/(RT) + \Delta S/R$, $\ln P_{eq} = -\Delta H/(RT) + \Delta S/R$) is the unit of P_{eq} , which means that the unit of P_{eq} is Pascal for $\ln P_{eq} = -\Delta H/(RT) + \Delta S/R$, while bar for $\ln(P_{eq}/P_0) = -\Delta H/(RT) + \Delta S/R$. The dissociation pressures at 500 °C calculated to be 22730.5-54069.9 MPa in the ZrCo-H system are unreasonably high.

R: Thank you for pointing out the error in the use of this equation. We have corrected the calculation and corresponding unit in the revised manuscript.

The contradiction between thermodynamic parameters and disproportionation kinetic curves is not yet precisely understood. Please carefully double check the thermodynamic parameters for dissociation process.

R: In the previous version, the plateau pressure P_{eq} was selected to be the left turning point of the plateau (*i.e.* P_L) in the PCT curve (as shown in yellow cycle in Fig. R1). The choice of this point is inappropriate because the plateau has a certain slope, especially for the PCT curves at 320 and 340 °C.

In the revised manuscript, we used the midpoint pressure of the plateau (*i.e.* P_M) (as shown in red cycle in Fig. R1) for Van't Hoff plots. The recalculated thermodynamic parameters are shown in Supplementary Table S10 (also given below). Based on these parameters, the dissociation pressures of our ZrCo samples at 500 °C should be 0.77-1.28 MPa, all higher than the disproportionation pressure of 0.7 MPa. In this case, when the hydrogenation disproportionation reaction occurs, the pressure will drop, which is consistent with the experiment. In the revised manuscript, we have replaced the old data with the new ones (Page 13, highlight).

Fig. R1. PCT curves of ZrCo alloy. The red and yellow circles represent the midpoints and left turning points (*i.e.* M and L) of the dehydrogenation plateaus at different temperatures.

Table R1. A summary of enthalpy (ΔH) and entropy (ΔS) values of ZrCo alloys.

Sample	ΔH (kJ mol ⁻¹)	ΔS (J mol ⁻¹ K ⁻¹)
ZC-67nm	95.7 ± 1.9	236.5 ± 3.2
ZC-336nm	97.3 ± 1.8	239.9 ± 3.2
ZC-621nm	97.5 ± 2.3	240.4 ± 3.9
ZC-1.3μm	97.8 ± 2.7	241.3 ± 4.6
ZC-42μm	98.0 ± 2.4	242.0 ± 4.1
ZC-94μm	98.2 ± 2.1	242.7 ± 3.6
ZC-177μm	99.7 ± 2.8	245.9 ± 4.8

Response to Reviewer #3:

Q1. The XRD patterns are shown as a function of 2 Theta, however, neither in the manuscript nor in the SI, the wavelength of the radiation is given. What X-ray source was used? Was there any monochromator applied, e.g. to avoid fluorescence of Co? On page 6 “The crystallite size of ZC-67nm was calculated to be 50.1 nm,” how was the average crystallite size calculated, which diffraction peaks have been used?

R: The XRD measurements were carried out using a Rigaku D/max 2500·X-ray diffractometer with Cu K α radiation ($\lambda = 0.15418$ ·nm). The X-ray diffractometer is equipped with a graphite monochromator to filter fluorescence. The detailed information has been provided in the characterization section (Page 20, highlight).

The crystallite size of ZC-67nm was calculated by the Rietveld refinement method, where all diffraction peaks were used to improve the accuracy. Briefly, XRD pattern was refined until a sufficiently good reliability factor ($R_{wp}=11.6\%$) was obtained. Then, we obtained the scaling factors and profile parameters of the XRD pattern. During this process, the effects of instrumental and strain broadening were eliminated. At last, we used the Lorentzian parameter (LX) obtained from the profile parameters to calculate the crystallite size by (Ref. *Appl. Phys. Lett.* 2009, 95, 191906):

$$d = \frac{3600\lambda}{\pi^2(LX)}$$

where d is the crystallite size and λ is the wavelength of the X-ray radiation.

In the revised manuscript, we clarified the calculation method (Page 6, highlight).

Q2. Response to reviewer #2, Q2: “grain-environment boundary density (ρ_{GE} , i.e. the specific surface area)” the authors connect the grain boundary density to the specific surface area, however, firstly the two quantities have different units. Specific surface area is given in square meters per gram, therefore, eq (29) in the SI is wrong. In the response to Q2, the authors exchanged “specific surface area” for “grain-environment boundary density”, however, SI page 33 is still different. What is correct now? What is the definition of a “grain-environment boundary density”? Did the authors measure the specific surface area by gas adsorption techniques?

R: I am sorry that we did not clarify that the “specific surface area” we used in the previous version is “volume-specific surface area”.

Generally, specific surface area can be expressed in two ways, *i.e.* volume-specific surface area (VSSA, surface area per unit volume (m^2/m^3)) and weight-specific surface area (SSA, surface area per unit mass (m^2/g)) (Refs. Neil Gibson et al. Volume-specific surface area by gas adsorption analysis with the BET method, *Characterization of Nanoparticles*, 2020, Chapter 4.1, 265; Nadiia Mameka et al. On the impact of capillarity for strength at the nanoscale. *Nation communications*, 2017, 8, 176). Given the ZC-67nm is a single crystal, the “grain-environment boundary density” is defined as the interface area between ZC-67nm and environment per unit volume, which is

equal to the volume-specific surface area.

To avoid this misunderstanding and simplify the expression, in the new revised version, the boundary densities of ZC-177 μm and ZC-67 nm are uniformly expressed as volumetric grain boundary density (ρ_b). Relevant discuss was provided in the revised main text (Page10, highlighted).

We conducted the N_2 absorption that shows the weight-specific surface areas of 6.9 and 0.3 $\text{m}^2 \text{g}^{-1}$ for ZC-67nm and ZC-177 μm , respectively (as shown in Fig. R2). According to the measured weight-specific surface area, a ρ_b of 0.052 nm^{-1} can be calculated for ZC-67nm (based on the mass density of ZrCo of 7.5 g/cm^3). This value is close to the modeling result of 0.088 nm^{-1} .

Fig. R2. Weight-specific surface area of ZC-67nm and ZC-177 μm measured by N_2 absorption.

REVIEWERS' COMMENTS:

Reviewer #1 (Remarks to the Author):

This manuscript reports the effects and mechanism of Ti, Cu, Y, La, Ce, Pr, Nd and Sm substitutions on the anti-disproportionation performance of ZrCo alloy by combining experiments and first-principles calculations. The authors have revised the manuscript according to the reviewers' comments. I hence recommend its publication in the current state.

Reviewer #3 (Remarks to the Author):

The authors added information concerning the XRD analysis. The term "specific surface area" has been replaced by "grain boundary density" resulting in more consistency.

The N₂ adsorption isotherms (Fig. R2) and their analyses are not really state-of-the-art, however, this is irrelevant for the present manuscript since the material is non-porous and surface is not important. In future work, the authors may consider DOI 10.1515/pac-2014-1117

In my opinion, the manuscript can be published.

REVIEWERS' COMMENTS:

Reviewer #1 (Remarks to the Author):

This manuscript reports the effects and mechanism of Ti, Cu, Y, La, Ce, Pr, Nd and Sm substitutions on the anti-disproportionation performance of ZrCo alloy by combining experiments and first-principles calculations. The authors have revised the manuscript according to the reviewers' comments. I hence recommend its publication in the current state.

R: We greatly appreciate the Reviewer for all the constructive comments and suggestions.

Reviewer #3 (Remarks to the Author):

The authors added information concerning the XRD analysis. The term "specific surface area" has been replaced by "grain boundary density" resulting in more consistency.

The N₂ adsorption isotherms (Fig. R2) and their analyses are not really state-of-the-art, however, this is irrelevant for the present manuscript since the material is non-porous and surface is not important. In future work, the authors may consider DOI 10.1515/pac-2014-1117

In my opinion, the manuscript can be published.

R: We highly appreciate the Reviewer for all valuable comments. We have studied the suggested literature, and we use it in our future work.